# Biofilm Formation by *Staphylococcus aureus* Clinical Isolates is Differentially Affected by Glucose and Sodium Chloride Supplemented Culture Media

**DOI:** 10.3390/jcm8111853

**Published:** 2019-11-02

**Authors:** Harshad Lade, Joon Hyun Park, Sung Hee Chung, In Hee Kim, Jung-Min Kim, Hwang-Soo Joo, Jae-Seok Kim

**Affiliations:** 1Department of Laboratory Medicine, Hallym University College of Medicine, Kangdong Sacred Heart Hospital, Seoul 05355, Korea; harshadlade@gmail.com (H.L.); joonhyunpark0617@gmail.com (J.H.P.); ssung929@gmail.com (S.H.C.); inhee6389@gmail.com (I.H.K.); jungmin510@gmail.com (J.-M.K.); 2Department of Pre-PharmMed, College of Natural Sciences, Duksung Women′s University, 33 Samyang-ro 144-gil, Dobong-gu, Seoul 01369, Korea; hwangsoojoo27@duksung.ac.kr

**Keywords:** *S. aureus*, methicillin-resistant *Staphylococcus aureus* (MRSA), biofilm, biofilm formation assay, glucose supplementation, SCC*mec* type, delta-toxin

## Abstract

*Staphylococcus aureus* (*S. aureus*) causes persistent biofilm-related infections. Biofilm formation by *S. aureus* is affected by the culture conditions and is associated with certain genotypic characteristics. Here, we show that glucose and sodium chloride (NaCl) supplementation of culture media, a common practice in studies of biofilms *in vitro*, influences both biofilm formation by 40 *S. aureus* clinical isolates (methicillin-resistant and methicillin-sensitive *S. aureus*) and causes variations in biofilm quantification. Methicillin-resistant strains formed more robust biofilms than methicillin-sensitive strains in tryptic soy broth (TSB). However, glucose supplementation in TSB greatly promoted and stabilized biofilm formation of all strains, while additional NaCl was less efficient in this respect and resulted in significant variation in biofilm measurements. In addition, we observed that the ST239-SCC*mec* (Staphylococcal Cassette Chromosome *mec*) type III lineage formed strong biofilms in TSB supplemented with glucose and NaCl. Links between biofilm formation and accessory gene regulator (*agr*) status, as assessed by δ-toxin production, and with mannitol fermentation were not found. Our results show that TSB supplemented with 1.0% glucose supports robust biofilm production and reproducible quantification of *S. aureus* biofilm formation *in vitro*, whereas additional NaCl results in major variations in measurements of biofilm formation.

## 1. Introduction

*Staphylococcus aureus* is one of the most common bacterial pathogens that colonize mammalian skin and/or mucosal membranes. *S. aureus* has major clinical importance both as a commensal bacterium and an opportunistic pathogen, causing a wide variety of infections, including simple soft skin infections, endocarditis, bacteremia, and severe pneumonia [1,2,3]. Biofilms are highly organized multicellular bacterial communities embedded within a complex matrix composed of polysaccharide, proteins, and/or extracellular DNA (eDNA), and contribute to reduced susceptibility to antimicrobials and persistence of biofilm-associated infections [4]. Biofilm formation by bacteria significantly contributes their survival in the host and has been considered as a key virulence factor responsible for serious chronic infections [5,6]. In clinical scenarios, such as indwelling medical devices or catheter-associated infections, the ability to form biofilms is crucial for *S. aureus* pathogenicity [7] and biofilm-associated *S. aureus* infections resist antimicrobial therapy and innate host defense mechanisms [8]. Furthermore, biofilm formation by *S. aureus* and antimicrobial resistance are functionally linked, as the biofilm phenotype expressed can be influenced by the acquisition of antimicrobial resistance [4,9]. Many foodborne diseases are also associated with biofilms and considered as an emergent public health concern [10].

The *S. aureus* responsible for biofilm-associated infections can have different genetic backgrounds and, therefore, express a different spectrum of virulence factors during infection [11]. For example, biofilm formation appears to be associated with several regulatory factors, including the *agr* quorum sensing system [12]. *Agr* upregulates expression of various toxins, including δ-toxin (a molecule with surfactant-like properties) [13], which contributes to *S. aureus* adhesion and biofilm development [14]. In addition, biofilm formation requires polysaccharide intercellular adhesin (PIA), which is produced and regulated by the intercellular adhesion (*ica*) ADCB operon [15]. The *ica*ADCB operon includes an N-acetylglucosamine transferase (*ica*A and *ica*B) [16], a deacetylase (*ica*D) [17], and a predicted exporter (*ica*C) [16]. Furthermore, some surface components, such as Staphylococcal protein A (*spa*), contribute to adhesion within biofilms [18]. Genotypic variation among *S. aureus* strains may additionally influence biofilm formation [19], but these associations are not consistently reported. In addition, the molecular epidemiology of methicillin-resistant *S. aureus* (MRSA) strains, as identified by staphylococcal cassette chromosome *mec* (SCC*mec*) typing, are also associated with biofilm formation [20].

A 96-well microtiter plate assay has been considered as the standard method for quantification of *S. aureus* biofilm formation *in vitro* [21]. This assay allows rapid, high-throughput screening of antibiofilm compounds and typically serves as a precursor to *in vivo* studies. Numerous studies have examined the ability of *S. aureus* clinical strains to form biofilms by this method [19,22,23,24,25]. These studies vary widely in assay conditions, the culture media used, supplementation of sugar or salt in culture media, and categorization of biofilm formation; however, several did not fully evaluate the relationship between biofilm formation and the molecular characteristics of strain relatedness and genetic heterogeneity.

Tryptic soy broth (TSB) is the most commonly used laboratory culture media for biofilm formation assay [26,27,28]. Furthermore, glucose is normally added in the TSB to promote *S. aureus* biofilm formation *in vitro* [29]; however, various glucose concentrations have been used in different studies. For example, TSB (which already contains 2.5 g/L glucose and 5.0 g/L NaCl) is often supplemented with additional glucose (0.25%, 0.4%, 0.5%, and 1.0%) [22,30,31,32,33,34]. In addition, brain heart infusion (BHI) (which already contains 2.0 g/L glucose) is also used as a culture media for *S. aureus* biofilm formation *in vitro* [35], with additional glucose supplementation (0.5%, 1.0%, and 1.5%) [36,37]. Lennox broth supplemented with different concentration of glucose (0 to 320 mg/dL in 20 mg/dL intervals) was also used to study the enhanced effect on biofilm formation by *S. aureus* [38]. Moreover, Lim et al. carried out a comprehensive study to elucidate the effect of NaCl (0.1% to 5.6%) on biofilm formation by *S. aureus* in TSB [39]. It is interesting to note that strain categorization, when based on biofilm production capacity, may not ensure reproducibility in a laboratory setting. Thus, there is likely to be considerable variability in biofilm formation data generated by different studies.

In medical biofilm study, sensitivity and assay reproducibility are crucially important for diagnosis and determining the efficacy of drugs given to prevent or reduce biofilm formation on indwelling medical devices [40]. The reproducible assay method provides researchers with evidence that data generated are objective and reliable and not influenced by bias. Therefore, even with complex systems simulating real medical devices, the culture media should be standardized to minimize variability in the biofilm formation assay. Here, we hypothesized that the culture media is the most influential parameter that results in data variability of the 96-well microtiter plate biofilm assay. Thus, we have (i) investigated the effects of glucose and NaCl on *S. aureus* biofilm formation, and (ii) undertaken genotypic and phenotypic characterization of *S. aureus* clinical isolates to determine any association with biofilm formation capacity.

## 2. Methods

### 2.1. Bacterial Strains and Culture Conditions

This study used 40 randomly selected *S. aureus* isolates (21 MRSA and 19 methicillin-sensitive *S. aureus*, MSSA) from a total of 360 strains that were recovered from blood samples between 2005 and 2014 at a university-affiliated hospital in Korea. Bacterial identification was performed using matrix-assisted laser desorption ionization time-of-flight (MALDI-TOF) mass spectrometry (MS) (Bruker Microflex LT, Bruker Daltonik GmbH, Bremen, Germany). Antimicrobial susceptibility of the strains was tested using the MicroScan WalkAway 96 plus system (Beckman Coulter, Atlanta, GA, USA). *S. aureus* strains were grown on mannitol salt agar (MSA) and/or mannitol salt broth (MSB) (KisanBio, Seoul, Korea). Stock cultures were stored in skimmed milk at −70 °C and revived onto blood agar plates (BAP) (Shinyang Diagnostics, Seoul, Korea) for 16–18 h at 37 °C under 5% CO_2_. For the biofilm formation assay, cultures were grown in TSB (Becton Dickinson, Franklin Lakes, NJ, USA) (TSB composition: tryptone (pancreatic digest of casein) 17.0 g/L, soytone (peptic digest of soybean) 3.0 g/L, glucose (= dextrose) 2.5 g/L, sodium chloride 5.0 g/L, and dipotassium phosphate 2.5 g/L).

### 2.2. Biofilm Formation Assay

Quantification of *S. aureus* biofilm formation was performed by 96-well microtiter plate assay as previously described, with the following modifications [21]. Briefly, *S. aureus* isolates were grown overnight at 37 °C on BAP under 5% CO_2._ A single colony from BAP was then cultured in TSB at 37 °C overnight and used to prepare a bacterial suspension equivalent of a 0.5 McFarland turbidity standard (MicroScan turbidity meter, Beckman Coulter, Inc., Atlanta, Georgia). The assay media was either TSB, or TSB supplemented with 0.5% or 1.0% d-(+)-glucose, 1.0% or 2.0% NaCl, or with both 1.0% glucose and 1.0% NaCl. Therefore, the total sugar or salt concentrations in TSB were either 7.5 g/L or 12.5 g/L d-(+)-glucose, 15.0 g/L or 25.0 g/L NaCl, or both 12.5 g/L glucose and 15.0 g/L NaCl. The culture media were inoculated to give a final bacterial concentration of 1 × 10^6^ CFU/mL and dispensed (200 μL/well) into wells of microtiter plates (Falcon, Corning Inc., Corning, NJ, USA) [22,41]. The uninoculated media served as a negative control. Biofilms were grown under stationary conditions for 24 h at 37 °C with 5% CO_2_. After incubation, the bacterial culture from each microtiter plate well was gently aspirated, the wells washed twice with 200 μL of phosphate-buffered saline (PBS, pH 7.4) to remove nonadherent bacteria, the adherent bacteria fixed by heating at 65 °C for 1 h and then stained with 150 μL of 0.1% (*w*/*v*) crystal violet (Sigma-Aldrich, St. Louis, MO, USA) for 5 min. Excess crystal violet stain was then discarded and the plates were washed twice with PBS (200 μL) to remove residual dye and then allowed to dry for 30 min at room temperature. The stain adherent biofilm was dissolved in 150 μL of 33% glacial acetic acid (*v*/*v*) per well for 30 min. Resulting biofilm formation was quantified by measuring the absorbance at 595 nm with a 96-well ELISA reader (Multiskan FC, Thermo Fisher Scientific, Waltham, MA, USA). Each assay consisted of eight replicates for each condition and performed on at least two occasions. For each assay, the outlier values were identified by Z-score method and the nonoutlier data used to calculate average biofilm formation. The negative control value was subtracted, and the data displayed as the mean absorbance ± standard deviation (SD). *S. aureus* strains were considered to form biofilms when the ABS_595_ value was three times the SD of mean absorbance of the negative control [42].

The *S. aureus* RN4220 strain, a well-characterized biofilm-producing strain, was included in the assay as a positive control for biofilm formation [22]. Biofilm formation was categorized into three groups using an arbitrarily established baseline (the biofilm formation by the positive control strain grown for 24 h in TSB). The *S. aureus* isolates that formed (i) biofilms ≥75% of the positive control were designated as strong biofilm producers, (ii) 25–75% biofilms of the positive control as moderate biofilm producers, and (iii) <25% biofilms of the positive control as weak biofilm producers [23].

### 2.3. SCCmec Typing and Detection of ica Genes

DNA for PCR amplification was extracted using the HiYield™ Genomic DNA Mini kit (Real Biotech Corp., Taipei, Taiwan) as per the manufacturer’s instructions. MRSA strains were subjected to SCC*mec* typing by discriminating the *mec* gene complex and the cassette chromosome recombinases (*ccr*) gene complex types, as previously described [43]. The presence of *icaA*, *icaD*, *icaB*, and *icaC* genes involved in biofilm formation of *S. aureus* was identified by PCR assay. Primers for typing *ica* gens were as described previously [44,45].

### 2.4. Multilocus Sequence Typing (MLST) and spa Analysis

All strains were screened for MLST based on seven housekeeping genes (*arcC*, *aroE*, *glpF*, *gmK*, *pta*, *tpi*, and *yqiL*) as described previously [46]. The *spa*-typing was performed using the protocol and primers published previously [47]. Sequence types (STs) and *spa* types were assigned using the BioNumerics software v.7.5 (Applied Math, Sint-Martens-Latem, Belgium).

### 2.5. Delta-Toxin Production

*Agr* functionality was assessed by the detection of δ-toxin production by MALDI-TOF MS [48]. The presence of the δ-toxin peak at *m*/*z* 3004 or its allelic variant at *m*/*z* 3034 was manually searched in the MALDI-TOF spectrum. The dysfunction of *agr* was defined as the absence of δ-toxin production [48].

### 2.6. Statistical Analysis

Statistical analyses were performed using SPSS software ver. 24.0 (SPSS Inc., IBM, Chicago, IL, USA) and Microsoft Excel 2016 (Microsoft, Redmond, WA, USA). Student’s *t*-test was used to compare the biofilm formation between the MRSA and MSSA strains. The level of significance was determined using one-way ANOVA with Tukey′s multiple comparison test. *P* ≤ 0.05 was considered significant.

## 3. Results

### 3.1. Effect of Glucose and NaCl Supplementation on Biofilm Formation

Prior to biofilm quantification, the cumulative bacterial growth consisting of the biofilm and planktonic cells in a 96-well microtiter plate was determined by measuring the absorbance at 595 nm (ABS_595_). This revealed robust growth for all 40 of the *S. aureus* clinical isolates when cultured in either TSB or TSB supplemented with different concentrations of glucose and NaCl (data not shown). Nevertheless, some differences in the level of growth were observed between strains and under the different culture media compositions. The biofilm formed on the bottom surfaces of microtiter plate wells was then assessed by crystal violet staining. Firstly, we established an arbitrary baseline for the categorization of biofilm formation by the *S. aureus* clinical isolates using the biofilm formed by a positive control strain, *S. aureus* RN4220, when grown in TSB alone. TSB supported growth and biofilm formation by all of the *S. aureus* strains when compared with that of the positive control strain (RN4220, ABS_595_ 0.31). On this basis, the strains were categorized as strong (ABS_595_ ≥ 0.54), moderate (ABS_595_ 0.39–0.54), or weak (ABS_595_ < 0.39) biofilm producers, as seen in Figure 1. As illustrated in Figure 1, distinct biofilm formation was observed between the *S. aureus* strains when grown in TSB. Notably, several MRSA strains showed significantly greater biofilm formation than the MSSA strains (*p* < 0.05). Among the 21 MRSA strains, 28.5% (*n* = 6/21) showed strong biofilm formation compared with only 5.2% (n = 1/19) of the MSSA strains. Furthermore, 19.0% (*n* = 4/21) of the MRSA and 5.2% (*n* = 1/19) of the MSSA strains formed moderate biofilms in TSB, while weak biofilms were formed by 52.3% (*n* = 11/21) and 89.4% (*n* = 17/19) of the MRSA and MSSA strains, respectively. Interestingly, although most of the MSSA strains formed weak biofilms in TSB, the *S. aureus* 14,669 strain formed a strong biofilm (ABS_595_ 1.17).

The addition of glucose to the culture media (TSB) strongly promoted biofilm formation by both the MRSA and MSSA strains. All 21 MRSA strains showed increased biofilm formation in TSB supplemented with 0.5% and 1.0% glucose, as seen in Appendix A and Figure 2, when compared with TSB alone. Additionally, glucose supplementation strongly promoted biofilm formation by most of the MSSA strains (0.5% glucose; 84.2%, *n* = 16/19 and 1.0% glucose; 94.7%, *n* = 18/19), whereas the other strains showed moderate biofilm formation (0.5% glucose; 15.7%, *n* = 3/19 and 1.0% glucose; 5.2%, *n* = 1/19). However, there was no statistically significant difference in biofilm formation between the MRSA and MSSA strains when they were grown in TSB supplemented with 0.5 or 1.0% glucose. These results show that glucose supplementation promotes biofilm formation of all of the strains tested, and suggest that the rate of biofilm formation is distinct for MRSA and MSSA strains.

Addition of 1.0% NaCl to TSB resulted in different levels of biofilm formation for both the MRSA and MSSA strains, as seen in Appendix A. For the MRSA strains, 57.1% (*n* = 12/21) showed robust biofilm formation, whereas robust biofilm was only seen for 15.7% (*n* = 3/19) of MSSA strains. When NaCl was increased to 2.0%, 80.9% (*n* = 17/21) of the MRSA and 52.6% (*n* = 10/19) of the MSSA strains formed strong biofilms, as seen in Appendix A. However, weak biofilm formation was also seen for 2.8% (*n* = 9/21) of the MRSA and 73.6% (*n* = 14/19) of the MSSA strains in 1.0% NaCl, and for 9.2% (*n* = 2/21) of the MRSA and 31.5% (*n* = 6/19) of the MSSA strains in 2.0% NaCl. Interestingly, the MSSA strain 14669 formed a strong biofilm in TSB supplemented with either 1.0% or 2.0% NaCl (ABS_595_ 1.74 and 3.43, respectively). Additionally, the MSSA strain 7403 also formed considerably strong biofilm in TSB supplemented with 2.0% NaCl (ABS_595_ 2.38), as seen in Appendix A. Notably, when compared with biofilm formation in TSB alone, the addition of 2.0% NaCl significantly promoted biofilm formation for the MRSA and MSSA strains; however, considerable variation in biofilms was observed. Our results show that average MRSA strains generated significantly stronger biofilms than MSSA strains when they were grown in TSB supplemented with 1.0% NaCl. The coefficient of variation (CV) was used to assess the variability of the data from the biofilm formation assay. This showed that the CV for the assay in TSB supplemented with NaCl ranged from 2.4% to 50% for 1.0% NaCl, and between 2.0% and 53.7% for 2.0% NaCl (data not shown), indicating poor reproducibility of biofilm formation in the presence of additional NaCl.

We also investigated the effects on biofilms of a combination of 1.0% glucose and 1.0% NaCl. The results show that 85.7% (*n* = 18/21) of the MRSA and 68.4% (*n* = 13/19) of the MSSA strains formed strong biofilms under these conditions, as seen in Appendix A. Moreover, 14.2% (*n* = 3/21) of the MRSA and 21.0% (*n* = 4/19) of the MSSA strains formed moderate biofilms when additional glucose and NaCl were both present, while two MSSA strains (*S. aureus* 10794 and *S. aureus* 9684) produced weak biofilms. However, the effect of adding both glucose and NaCl to TSB on biofilm formation by the MRSA and MSSA strains was not significantly different.

### 3.2. SCCmec Types and Biofilm Formation

Next, we investigated if different SCC*mec* types influenced biofilm formation by the MRSA strains when grown in different culture media compositions. Molecular epidemiology analysis of MRSA strains revealed the following SCC*mec* types: 38.0% were SCC*mec* type II (*n* = 8), 33.3% (*n* = 7) SCC*mec* type IV, and 28.5%, (*n* = 6) SCC*mec* type III, as seen in Table 1. Notably, only 66.6% (*n* = 4) of the SCC*mec* type III strains formed strong biofilms in TSB, while two strains (*S. aureus* 2065 and *S. aureus* 2096) showed weak biofilm formation. All SCC*mec* type III strains formed strong biofilms in glucose (0.5% and 1.0%), NaCl (1.0% and 2.0%), and glucose plus NaCl (1.0% each) supplemented TSB, as seen in Figure 3. Similarly, although the SCC*mec* type IV strains showed weak biofilm formation in TSB, they showed strong biofilm formation in TSB supplemented with glucose, while their biofilm formation differed when they were grown in TSB supplemented with NaCl and TSB supplemented with glucose plus NaCl. No statistically significant difference in biofilm formation by different SCC*mec* type strains was observed when they were cultured in TSB supplemented with 1.0% glucose or 2.0% NaCl, although supplementation with 2.0% NaCl resulted in wide differences in biofilm formation between strains.

### 3.3. Multilocus Sequence Typing (MLST) and spa Analysis

The evolutionary and genetic diversity of the *S. aureus* strains was analyzed by MLST and revealed a total of 15 distinct STs, as seen in Table 1. The MSSA strains were more diverse than MRSA strains with a higher number (*n* = 12) of STs (ST1, ST6, ST8, ST30, ST45, ST59, ST72, ST97, ST188, ST513, ST1821, and ST1970). Of these, the most common were ST188 (15.7%, *n* = 3), ST1, ST8, ST59, ST72, and ST97 (each 10.5%, *n* = 2). Among the 21 MRSA strains, MLST analysis identified five different STs (ST1, ST5, ST72, ST239, and ST686), with the most common being ST5 (33.3%, *n* = 7), ST72 (23.8%, *n* = 5), and ST239 (28.5%, *n* = 6). A new ST was identified for MRSA strain 2102 and designated ST686 and deposited in the online database (https://pubmlst.org/saureus/). The genetic variation analysis among the *S. aureus* strains showed a total of 22 *spa* types, as seen in Table 1. Among the MRSA strains, *spa* t037 (28.5%, *n* = 6) was the most dominant, followed by t9353 (19.0%, *n* = 4), t324 (14.2%, *n* = 3), and t286 and t601 (each 9.5%, *n* = 2). The following *spa* types were seen only once: t148, t2460, t5607, and t664 (each 4.7%, *n* = 1). Of the MSSA strains, 14 *spa* types were identified, among which types t008, t126, and t127 (10.5%, *n* = 2), and t189 (15.7%, *n* = 3) were most frequently seen. Each of the remaining *spa* types were identified in 10 MSSA strains.

### 3.4. The icaADCB Genes and Mannitol Fermentation

Our results showed that all of the genes encoded by the *ica*ADCB operon were present in all 40 *S. aureus* strains. Therefore, an association between *S. aureus* biofilm formation and the prevalence of *ica* genes could not be evaluated. Results of growth on mannitol salt agar identified three MRSA (*S. aureus* 7875, *S. aureus* 9770, and *S. aureus* 2096) and two MSSA (*S. aureus* 6280 and *S. aureus* 4308) strains as mannitol-negative, as seen in Table 1. All of the mannitol-negative MRSA and MSSA strains formed weak biofilms in TSB. Additionally, some of the mannitol-positive MRSA (44.4%, *n* = 8/18) and MSSA (88.2%, *n* = 15/17) strains also showed weak biofilm formation in TSB. All of the mannitol-negative and mannitol-positive MRSA strains (*n* = 21) showed strong biofilm formation when the media was supplemented with glucose (0.5% and 1.0%).

### 3.5. Agr Status and Biofilm Formation

MALDI-TOF MS confirmed the presence of δ-toxin (*m*/*z* 3004 and/or its allelic variant at *m*/*z* 3034) for 36 isolates; MRSA strains (*n* = 18) and MSSA strains (*n* = 18), as seen in Table 1. Two of the *agr* dysfunctional strains (MRSA, *S. aureus* 12779 and MSSA, *S. aureus* 6280), identified by the lack of δ-toxin, formed weak biofilms, while two other MRSA (*S. aureus* 11111 and *S. aureus* 9291) strains formed moderate biofilms in TSB, as seen in Figure 1. All strains with a dysfunctional *agr* showed strong biofilm formation in TSB supplemented with 0.5% and 1.0% glucose, as seen in Appendix A and Figure 2, weak biofilm formation in TSB supplemented with 1.0% NaCl, as seen in Appendix A, different levels of biofilm formation in TSB supplemented 2.0% NaCl, as seen in Appendix A, and strong to moderate biofilm formation in TSB supplemented with 1.0% glucose and 1.0% NaCl, as seen in Appendix A. Similarly, *agr* functional strains also formed strong to moderate biofilms in glucose supplemented TSB, distinct biofilms in NaCl supplemented TSB, and strong to moderate biofilms in TSB supplemented with both glucose and NaCl. Collectively, these data show that there was no significant difference in biofilm formation by strains with functional or dysfunctional *agr* under the tested media compositions.

## 4. Discussion

TSB, a less nutrient-rich laboratory medium which is commonly used for biofilm formation studies [26,27,28] was shown to support the growth of *S. aureus*, but with insufficient biofilm formation [22]. However, supplementing TSB with 1.0% glucose resulted in promoted biofilm formation of 40 *S. aureus* clinical strains with different genetic backgrounds *in vitro*, as seen in Figure 2. We also found that the addition of glucose resulted in more consistent biofilm quantification when using a 96-well microtiter plate assay, as less variation (i.e., lower SD) was seen between replicates and between assays. The evidence of glucose-induced biofilm formation for *S. aureus* has been observed [10], and subsequently confirmed in several studies [14,19,22,49]. We did not determine pH changes in the culture media, but glucose represses the *agr* quorum sensing system [29] due to the excretion of short-chain fatty acids that result from glucose metabolism, which lowers the pH of the surrounding medium [50]. An acidic pH represses extracellular protease production [51], stimulates the association of biofilm matrix proteins on cell surfaces [52,53], and instigates functional amyloid assembly [54], which promotes biofilm formation.

Addition of NaCl to TSB also enhanced the biofilm formation of several *S. aureus* strains tested; however, considerable differences in biofilm formation and experimental data was observed. This is likely due to the loose attachment of *S. aureus* biofilms to the well surface when excess NaCl is present. We also observed that NaCl (1.0% and 2.0%) induced biofilm formation more effectively for MRSA (57.1% and 80.9%) than MSSA strains (15.7% and 52.6%). These results are contrary to previously published work, where biofilm formation in the presence of additional NaCl was more efficient for MSSA strains over MRSA strains [22]. However, this discrepancy could be explained by the use of different NaCl concentrations between the different studies, as the study by Sugimoto et al. used 4.0% NaCl [22], which is twice the salt concentration we tested. As described, NaCl activates expression of the *ica*ADBC operon in *S. aureus*, resulting in PIA production [14], which is required for biofilm formation and stability. However, although NaCl-induced activation of transcription from the *icaADCB* operon is stronger for MRSA than MSSA strains, it does not result in PIA production in MRSA strains [14,18]. Our results showing poor reproducibility and large differences in CV (2.0–53.7%) between replicates and between assays, when biofilm formation is determined by the 96-well microtiter plate assay in the presence of additional NaCl, will be significant for the development of a consistent assay method. In addition, TSB supplemented with 1.0% glucose produced robust and consistent biofilms, leading to greater assay reproducibility, and thus should result in improved quantification of *S. aureus* biofilms *in vitro*.

We also investigated the association between genotypic and phenotypic characteristics of *S. aureus* strains with biofilm formation when varying concentrations of glucose and NaCl were added to the culture media. Our results suggest that MRSA strains with SCC*mec* type III form strong biofilm in glucose and NaCl supplemented TSB, whereas SCC*mec* type I and II strains formed strong biofilms in the presence of glucose and differed when grown in NaCl supplemented TSB. The most prevalent hospital associated (HA) MRSA lineage that carries the SCC*mec* type III is MRSA-ST239 [55] and recognized as multidrug resistance across the globe [56]. In the present study, we found that 28.5% (*n* = 6/21) of the MRSA strains tested were of the ST239 lineage (typically referred as HA strains), and of these, the ST5-SCC*mec* II and ST72 lineages predominated. These finding are consistent with a previous report that showed the prevalence of ST239 (53.0%, *n* = 100/188) and ST5 (34.0%, *n* = 63/188) MRSA clones in Korean hospitals during 2001–2004 [57]. In addition, the ST5 [58] and ST239 [59] lineages have been identified as the dominant clones in clinical specimens from other geographical regions. We detected *ica*ADCB genes in all the 40 *S. aureus* strains tested. However, previous studies on biofilm formation by *S. aureus* strains did not identify all of the *ica* genes, and the *icaA* and *icaD* genes were more frequently observed [24,60,61]. Regardless of mannitol fermentation, different levels of biofilm formation were observed among strains grown in TSB supplemented with 1.0% and 2.0% NaCl as well as in TSB supplemented with combination of 1.0% glucose and 1.0% NaCl. In our study, an association between mannitol fermentation and biofilm formation was not observed in the *S. aureus* strains tested.

We found no association between *agr* status (as assessed by δ-toxin production) and biofilm formation under any of the assay conditions. However, previous studies suggest that *agr* dysfunction is associated with increased biofilm formation in *S. aureus* [17,62]. A limited number of δ-toxin negative strains may have impaired the ability to establish association between robust biofilm formation and *agr* dysfunction. Here, to minimize the sample size bias, the *S. aureus* clinical isolates were randomly selected from a total of 360 strains recovered between 2005 and 2014 and were blind to the biofilm formation ability. Additionally, we used *S. aureus* RN4220 as a positive-control biofilm-forming strain to ensure comparability between results.

In conclusion, our findings show that biofilm phenotype of *S. aureus* clinical strains can vary considerably depending on the composition of the culture media. In particular, glucose added to TSB promoted robust biofilm formation and resulted in improved assay results, while the addition of NaCl resulted in less biofilm formation and significant variation in biofilm quantification when using a 96-well microtiter plate assay. Associations between mannitol fermentation and *agr* functionality with biofilm formation were not observed under all of the tested culture media compositions. This study shows that TSB supplemented with 1.0% glucose is the most appropriate culture medium for promoting biofilm formation and results in higher assay reproducibility when quantifying *S. aureus* biofilm formation *in vitro*.

## Figures and Tables

**Figure 1 jcm-08-01853-f001:**
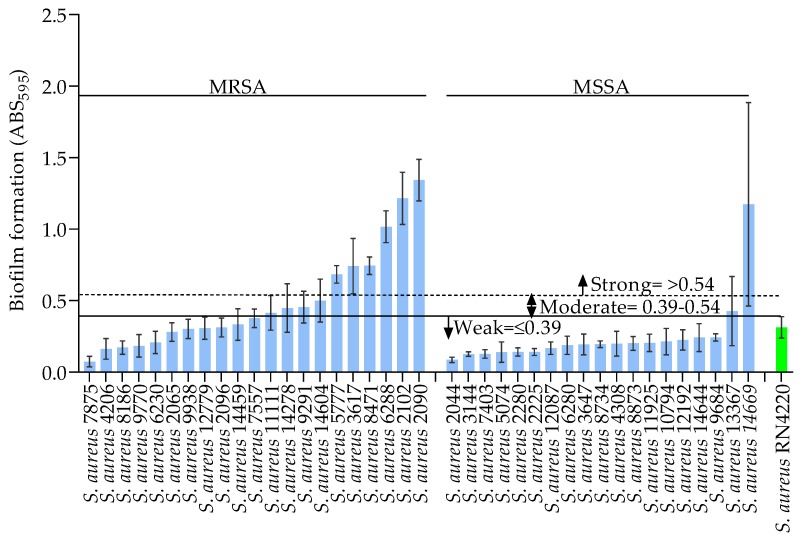
Biofilm formation (ABS_595_) of MRSA and MSSA strains grown in TSB. The green bar shows results for the positive control strain RN4220. The data show the mean, and error bars represent the standard deviation within the replicates, for at least two independent experiments.

**Figure 2 jcm-08-01853-f002:**
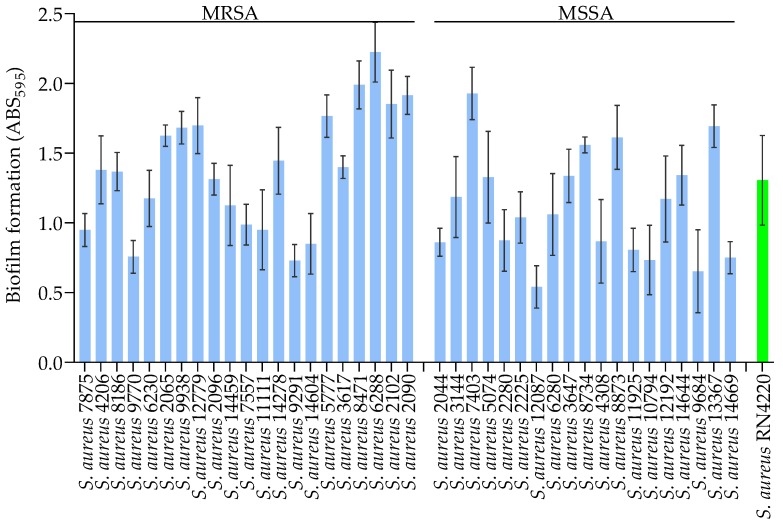
Biofilm formation (ABS_595_) of MRSA and MSSA strains grown in TSB supplemented with 1.0% glucose. The green bar shows results for the positive control strain RN4220. The data show the mean, and error bars represent the standard deviation within the replicates, for at least two independent experiments.

**Figure 3 jcm-08-01853-f003:**
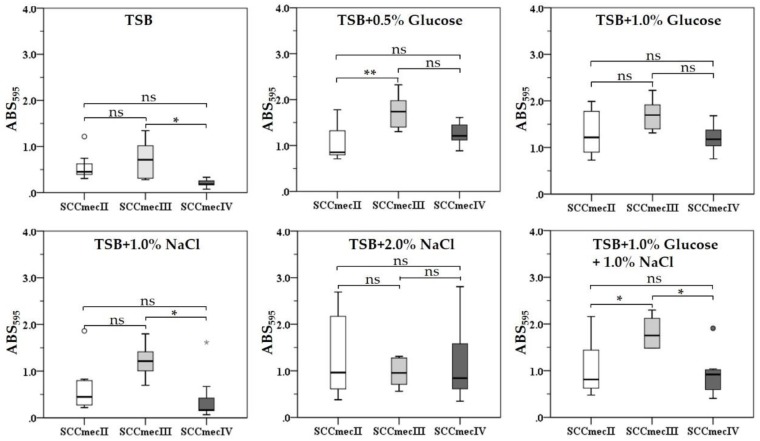
Biofilm formation by MRSA strains with distinct SCC*mec* types (SCC*mec* II, *n* = 8; SCC*mec* III, *n* = 6; and SCC*mec* IV, *n* = 7) grown in different culture media compositions. Box plot shows the median (horizontal thick blank line) of biofilms formed by MRSA strains. The lower and higher edges of the box represent the 25th and 75th percentiles, respectively. Box plot whiskers represent the 90th and 10th percentiles. *P* values were determined using Student′s *t*-test and asterisks indicate statistical significance (* = *p* ≤ 0.05, ** = *p* < 0.01). (^ns^ = *p* > 0.05, not statistically significant).

**Table 1 jcm-08-01853-t001:** Phenotypic and genotypic characteristics of the *S. aureus* strains used in this study.

*S. aureus* Strains	SCC*mec* Type (Only for MRSA)	MLST ST	*spa* Types	Mannitol Fermentation	δ-toxin (*m*/*z* 3004 and 3034*)
**MRSA**					
7875	IV	72	t664	N	+
4206	IV	1	t286	P	+*
8186	IV	72	t324	P	+
9770	IV	72	t148	N	+
6230	IV	72	t324	P	+
2065	III	239	t037	P	+
9938	IV	1	t286	P	+*
12779	II	5	t2460	P	−
2096	III	239	t037	N	+
14459	IV	72	t324	P	+
7557	II	5	t9353	P	+
11111	II	5	t601	P	−
14278	II	5	t9353	P	+
9291	II	5	t601	P	−
14604	II	5	t9353	P	+
5777	III	239	t037	P	+
3617	III	239	t037	P	+
8471	II	5	t9353	P	+
6288	III	239	t037	P	+
2102	II	686	t5607	P	+
2090	III	239	t037	P	+
**MSSA**					
2044	O	97	t267	P	+
3144	O	513	t164	P	+
7403	O	97	t9353	P	+
5074	O	45	t1460	P	+
2280	O	188	t189	P	+
2225	O	1	t127	P	+*
12087	O	188	t189	P	+
6280	O	1	t127	N	−
3647	O	30	t318	P	+
8734	O	188	t189	P	+
4308	O	59	t1950	N	+*
8873	O	72	t126	P	+
11925	O	6	t701	P	+
10794	O	1970	t065	P	+
12192	O	8	t008	P	+
14644	O	72	t126	P	+
9684	O	59	t1151	P	+*
13367	O	8	t008	P	+
14669	O	1821	t5554	P	+

MLST: Multilocus sequence typing, CC: Clonal complexes, ST: Sequence type, *spa*: Staphylococcal protein A, O: Nontypable, P: Positive, N: Negative, +: Present, −: Absent, *m*/*z* 3034*: Allelic variant of δ-toxin.

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
