# Peer review of "Biofilm Formation by Staphylococcus aureus Clinical Isolates is Differentially Affected by Glucose and Sodium Chloride Supplemented Culture Media"

_jcm, 2019, doi:10.3390/jcm8111853_

Round 1

Reviewer 1 Report

In the manuscript entitled ‘’Biofilm formation by staphylococcus aureus clinical isolates is differently affected by glucose and sodium chloride supplemented culture media’’, the authors show that biofilms are impacted by culture conditions such as glucose, NaCl or both. They provide necessary controls to their experimental designs, the material and method section is clear and enough descriptive. Plus they provide here a complete investigation with more than 40 different strains sensitive or not to the antibiotics.

However, they introduce the fact (in lines 71-73) that another study already showed that NaCl impacted biofilms (Lim et al, ref. 37). To be innovative, the authors aims among other things to provide the evidence that glucose is also a modulator of biofilm formation. They conclude in the discussion section that their findings show that biofilm formation depends on media composition ... In particular, glucose added to TSB promoted robust biofilm formation (lines 386-390).

Therefore, this is my main concern. I noticed two previous studies that have already shown that in vitro biofilm formations by bacteria including the staphylococcus aureus is enhanced when glucose is added in cultures (Jahid IK et al., Journal of Food Protection, 2013; PMID ?) (Waldrop R. et al., Clin. Orthop. Relat. Res., 2014; PMCID: PMC4182383; PMID: 24599648).

I recognize the fact that I’m no expert on the matter. But, I strongly recommend the authors:

to add these TWO references, especially the second study on aureus; and to discuss in the introduction and discussion section why their results are really innovative and bring something new on the table.

described a nouvel and potential therapeutic endeavor against the aggressive form of leishmaniasis, visceral leishmaniasis (VL). The authors underlies some critical events regarding the immunological status of the infected host that could be a vital precursor for designing an effective vaccine. In the manuscript, the authors investigated the abundance of two essential dendritic cells (DCs) subsets, Lymphoid and Myeloid, during the establishment of murine experimental VL. Thus their impact on naive T cells priming to either Th1 or Th2 phenotypes was diligently evaluated. In addition, the authors point out an important downstream regulation of some immunological accessory cell surface molecules, specifically OX40L and SEMA 4A. The findings of this study is undoubtedly compelling. However, there’re still some limitations and questions regarding the study’s experimental design including the manuscript itself structure-wise.

Major strength points:

  The experimental model used for this study (L.Donovani infected Balb/c mice) is consistent in the most of studies regarding VL. The authors successfully included controls for each set of experiment including the use of miltefosine (FDA approved treatment leishmaniasis). The data show significance and promising perspectives in a sequential manner.   Limitations & weaknesses:

Reviewer 2 Report

This is a standard article as a lot presented before in the literature. It is a potentially interesting research with a large body of data whose impact for diagnosis and management of biofilm-associated infections has not been sufficiently highlighted. The authors should highlight more the importance of the development of a consistent (sensitive and reproducible) biofilm formation assay method that improves quantifications of Staphylococcus aureus biofilms in vitro. A method of these conditions is crucial for determining the efficacy of drugs given to prevent or reduce biofilm formation on indwelling medical devices. The manuscript is well written and Tables and Figures support the conclusions included. However there are some aspects that should be corrected before acceptance for publication in Journal of Clinical Medicine.

Specific comments

- Line 65 and others: Replace or delete 'Importantly' where it appears.

- Line 65: Tryptic soy broth (TSB)......

- Line 89: using Matrix Assisted Laser Desorption Ionization Time-of-Flight (MALDI-TOF) mass spectrometry (MS) (........).

- Lines 94-95: in TSB…..

- Lines 144-145: MALDI-TOF MS.

- Line 170: the MSSA strains (

- Lines 220-222: It is very confusing and difficult to understand. In my opinion your idea is perfectly understood by eliminating the tail of the sentence, that is, 'to when they were added individually'.

- Lines 244-247: Delete 'However' and replace 'whereas' by 'in spite of’ or 'although'.

-Line 257: …distinct STs (

- Lines 264-269: In my opinion spa should go in italic.

- Lines 285-286: MALDI-TOF MS confirmed……

Reviewer 3 Report

The authors described two main conclusions in this manuscript. One is that TSB supplemented with 1% glucose is the most suitable medium for the quantification of biofilm formation. Another is that SCCmec type III strains formed stronger biofilms than other SCCmec genotype strains. However, biofilm formations in TSB supplemented with 1% glucose were not significantly different between SCCmec type III strains and other genotype strains. This seems to be contradiction. Term usage is inconsistent. The authors used “biofilm biomass” in section 3.1, and “biofilm formation” in other section. How different were biofilm biomass and biofilm formation? The authors used “strong biofilms (and weak biofilms)”. Does this mean formation of a large amount of biofilms or formation of physically solid biofilms? Discussion section is too long and complicated. 1, Table 2, Figure 2, and supplemental Figures were made by the same data of biofilm formation experiments. The duplications in data presentation should be avoided. Table 2: Some data are unnecessary. For example, ica genes were positive for all strains examined. Table S1 is no sense, because it can be available from the MLST database. 9, line 279-283: What is conclusion derived from these results?

Round 2

Reviewer 3 Report

The new version of manuscript is well revised according the comments given by the reviewers. However, there are several comments as shown below.

Comments:

Abstract: In the sentence “Links between biofilm formation and accessory gene regulator (agr) status, ---, were not found.”, “mannitol formation” should be added. In Discussion section, there are repetitive descriptions (for example, p. 9, line 296-298, p. 9, line 300-302, and p. 10, line 323-325). p. 10, line 341-342: The reference [62] seems to be wrong. This paper does not include the data of ica genes. And this sentence seems to mean that mRNA expression levels of ica are influenced by environmental condition, but not the existence of ica genes in genome. p. 11, line 358: “ica gene prevalence” should be deleted, because comparison between ica-positive strains and ica-negative strains has not been examined in the present study. Figure 2: What do you mean an arrow in the bar of S. aureus 3647? Table 1 should be compressed into one page.
